# Double Rounding Quantization for Flexible Deep Neural Network Compression

## Abstract

Model quantization is widely applied for compression and acceleration of deep neural networks, due to its simplification and adaptability. The quantization bit-width is typically predefined for quantizing a given neural network. However, the bit-width settings vary in different hardware and transmission demands, which will induce considerable training and storage costs. Therefore, the scheme of once-joint training for multiple bit-widths (multi-bit) is proposed to address this issue. In this paper, we propose a *Double Rounding* quantization method that can save the highest bit-width model instead of the full-precision counterpart and fully exploits the representation value range. Nevertheless, the performance during once-joint training degrades significantly due to inconsistent gradients between high-bit and low-bit quantization. To tackle this problem, we set the learning rate of multi-bit to proper values in an adaptive manner during training. We also apply our method for mixed-precision super-net and provide a novel training strategy with weighted probability. Experimental results demonstrate the proposed method outperforms the SOTA once-joint quantization-aware methods on ImageNet datasets. The code will be available soon.

## 1 Introduction

In recent years, with the popularity of mobile terminals and smart devices, more and more researchers have attracted attention to the field of model compression due to the limitations of computing resources and storage costs. The main popular methods of model compression include model pruning (Li et al., 2016), knowledge distillation (Gou et al., 2021) and model quantization (Zhou et al., 2016; Esser et al., 2019). Among them, model quantization has gained significant prominence within the industry. By mapping continuous floating-point values to discrete integer values in model representations, quantization offers the remarkable advantage of substantially reducing storage requirements and minimizing computational latency without altering the underlying network architecture.

Typically, given a pre-trained model, the configuration of quantization bit-width is pre-defined for a specific application scenario, and subsequently, the quantized model undergoes retraining or fine-tuning to mitigate the decline in accuracy. Nonetheless, when dealing with diverse application scenarios, this often requires the repetition of processes using the same model. In addressing this challenge, solutions involving the simultaneous training of multi-bit (Jin et al., 2020; Yu et al., 2021; Xu et al., 2022) have been proposed. These approaches involve weight parameter sharing between low-precision models and high-precision counterparts, enabling dynamic bit-width switching during inference. However, it is important to note that the bit-switching process can introduce significant performance degradation. Prior methods either store the float-pointing parameters to avoid this problem or abandon some integer values by replacing *rounding* with *floor* to satisfy the bit-switching. The latter is a rigorous decline, especially when 2-bit quantization is involved. In this paper, we propose a novel bit-switching method called *Double Rounding*, which utilizes the rounding operation twice compared to the conventional quantization process. In this way, not only the bit-switching is lossless but also can store the highest bit-width model instead of full-precision.

For the scheme of multi-bit joint quantization, the lower bit-width is part of the higher bit-width to fulfill storage constraints. For example, the first four bits of the 8-bit weights are identical to the 4-bit weights. As a result, the quantization intervals of different bit-width models adhere to

a strict logarithmic relationship to meet the requirements of lossless bit-switching. However, the optimization of joint training multi-bit still faces a great challenge due to the optimal quantization intervals of high-bit and low-bit are significantly different natively. We empirically find the quantization interval's gradients of different bit-widths exhibit distinct magnitudes during training. Based on the above findings, we propose a method called *Multi-LR*. The learning rates of different bit-width models should be set adaptively during training.

To enhance the versatility of the *Double Rounding* method, we introduce a more generally mixed-precision quantization approach. Unlike traditional mixed-precision quantization, which initially defines a bit-width configuration for various layers and subsequently trains the quantized model based on this configuration, our approach offers a mixed-precision super-net with *Weighted Probability*. Given this super-net, one can readily obtain an optimal subnetwork based on specified parameter size or FLOPs requirements without any additional training or fine-tuning. During the training of mixed-precision super-net, layers with higher sensitivity are set to higher bit-width and vice versa. Specifically, the Hessian trace (Dong et al., 2020) is utilized to measure the sensitivity of each layer, and we use the Integer Linear Programming (Yao et al., 2021) to find the Pareto Frontier optimal solution of the bit-width configurations.

In conclusion, our main contributions can be described as:

- *Double Rounding*: A novel multi-bit quantization method that preserves integer-weight parameter storage without diminishing the representation values.
- *Multi-LR*: A innovative training strategy for multi-bit models effectively narrows the training convergence gap between high-precision and low-precision models. In contrast to the original strategy, our approach enhances the performance of low-precision models without compromising the high-precision model's performance.
- *Weighted Probability*: A novel training strategy for mixed-precision super-net. The access probability of bit-width for each layer is determined based on the layer's sensitivity, aligning with the subnetwork's decision-making process during inference.
- Experimental results on ImageNet datasets show that the proposed method outperforms the state-of-the-art methods under different mainstream network architectures.

## 2 RELATED WORKS

**Model Quantization** Generally speaking, model compression is a broader term that includes model pruning (Molchanov et al., 2019), knowledge distillation (Cho & Hariharan, 2019), matrix decomposition (Denil et al., 2013), and model quantization (Choi et al., 2018; Bhalgat et al., 2020). Model quantization is a technique that refers to converting floating-point values into integers to reduce model storage and accelerate model inference. It is commonly divided into uniform quantization and non-uniform quantization (Jacob et al., 2018). Besides, encompasses a range of methods, such as binarized, ternary neural networks (Hubara et al., 2016; Courbariaux et al., 2015), and mixed precision networks (Wu et al., 2018). In addition, some studies have explored the application of model quantization in different tasks, such as object detection (Chen et al., 2021), speech recognition (Shangguan et al., 2019), and natural language processing (Zafrir et al., 2019). For different tasks, it is necessary to select appropriate quantization methods according to the characteristics of the data and the model architecture.

**Once-for-All in Model Quantization** One-shot quantization-aware training, also known as Once-For-All (OFA), entails the training of a single model with multi-bit widths. This approach enables adaptive and lossless bit-width switching, accommodating varying computing resources and storage constraints, all without the need for retraining. It involves the sharing of the highest-bit parameter among different bit-width configurations. Previous works closely related to this approach include AdaBits (Jin et al., 2020), Any-precision (Yu et al., 2021), Bit-Mixer (Bulat & Tzimiropoulos, 2021) and MultiQuant (Xu et al., 2022). AdaBits is the first work to consider adaptive model compression from a quantization perspective, but it encounters convergence issues with 2-bit quantization on ResNet50 (He et al., 2015). Bit-Mixer addresses this problem by using the LSQ (Esser et al., 2019) quantization method, but it discards the lowest quantization value, resulting in a performance loss. Multi-bit joint quantization can also be viewed as a multi-objective optimization problem. Any-precision and MultiQuant combine knowledge distillation techniques to improve overall model

performance. Similar to our approach, AdaBits and Bit-Mixer are capable of saving an 8-bit model, while other methods rely on 32-bit models for bit switching. Additionally, Bit-Mixer and MultiQuant implement a layer-adaptive mixed-precision model, but MultiQuant needs to fine-tune 300 epochs to idea performance. We also propose a training strategy with *Weighted Probability*, which further enhances the performance of the mixed-precision model. Furthermore, we implement a decision-making method for selecting the optimal bit-width combinations of different layers of the model. It is emphasized that, unlike the NAS approach (Shen et al., 2021), which focuses on changing the network architecture, such as depth, kernel size, or the number of channels, our approach achieves model's OFA solely through the quantization technique without altering the model architecture. This leads to a reduction in training time by approximately 10% (Du et al., 2020) compared to independent quantization training.

## 3 METHODOLOGY

In this section, we begin by providing a detailed introduction to our novel multi-bit quantization method called *Double Rounding*. Subsequently, we delve into the implementation methods for multi-bit joint training and present a training strategy incorporating multiple learning rates (*Multi-LR*). Finally, we explore the adaptability of our method by applying *Double Rounding* to mixed precision and introduce a training strategy featuring *Weighted Probability*, which enhances the performance of mixed precision models.

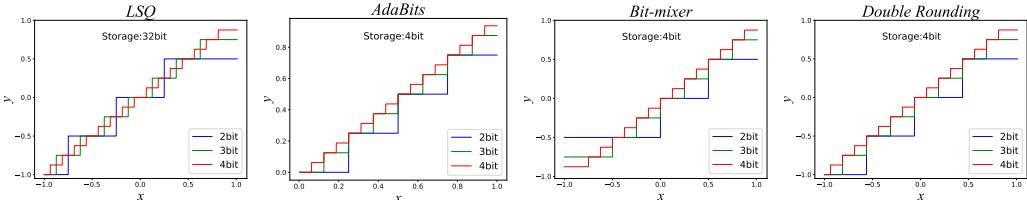

Figure 1: Comparison of four multi-bit quantization schemes:(from left to right) used in *LSQ* (Esser et al., 2019), *AdaBits* (Jin et al., 2020), *Bit-Mixer* (Bulat & Tzimiropoulos, 2021) and *Double Rounding*(Ours). Note that our *Double Rounding* differs from *Adabits* in that our quantization values can represent with $[-1, 1]$ not need to be fixed $[0, 1]$. In all cases $y = dequant(quant(x))$.

### 3.1 *Double Rounding*

In single-bit quantization methods using Quantization-Aware Training (QAT) (Jacob et al., 2017), given a pre-trained model, a pseudo-quantization node is inserted into each layer of the model during training. This pseudo-quantization node comprises two operations: the quantization operation, denoted as $quant(x)$, which maps floating-point (FP32) values to lower-bit integer values, and the dequantization operation, denoted as $dequant(x)$, which restores the quantized integer value to its original floating-point representation. It is employed to simulate the quantization error incurred when compressing continuous float values into discrete float values. As quantization involves a non-differentiable $Rounding$ operation, Straight-Through Estimator (STE) (Bengio et al., 2013) is commonly used to enable back-propagation and handle the non-differentiability. A general formulation for uniform quantization process (Li et al., 2021) is as follows:

$$\widetilde{\boldsymbol{W}} = \text{clip}\left(\left\lfloor \frac{\boldsymbol{W}}{\boldsymbol{s}} \right\rceil + \boldsymbol{z}, -2^{b-1}, 2^{b-1} - 1\right), \quad \widehat{\boldsymbol{W}} = (\widetilde{\boldsymbol{W}} - \boldsymbol{z}) \times \boldsymbol{s} \tag{1}$$

where the symbol $\lfloor . \rceil$ denotes the $Rounding$ function, clip$(x, low, upper)$ produces a tensor where entries below $low$ are set to $low$ and entries above $upper$ are set to $upper$. $b \in \{1, 2, \ldots, 8\}$ denotes the quantization level (or bit-width), $\boldsymbol{s} \in \mathbb{R}$ and $\boldsymbol{z} \in \mathbb{Z}$ represents the quantization *scale* (or interval) and *zero-point* associated with each $b$, respectively. $\boldsymbol{W}$ represents the model's weights, $\widetilde{\boldsymbol{W}}$ signifies the quantized integer weights, and $\widehat{\boldsymbol{W}}$ represents the dequantized floating-point weights.

However, in the case of multi-bit quantization, bit-switching can result in accuracy loss, especially when transitioning from higher bit-widths to lower ones (*e.g.* from 8-bit to 2-bit). To mitigate this

loss, prior works have employed two strategies. One approach involves switching from a floating-point model (32-bit) to a lower-bit model every time by using multiple learnable quantization scales. The other strategy substitutes the $Rounding$ operation with the $Floor$ operation to minimize the error during bit-switching. Nevertheless, the former approach cannot save the model as a quantized integer representation, which may not be suitable for devices with stringent storage constraints. The latter approach, by using the $Floor$ operation, sacrifices some integer values during the bit-switching process, which can be detrimental for multi-bit quantized models, particularly those with lower bit-widths (*e.g.* 2-bit). In contrast, we propose an innovative multi-bit quantization method called *Double Rounding*. This method overcomes these limitations by employing a twice $rounding$ operation. It allows the model to be saved in the highest-bit representation instead of full-precision, facilitating seamless switching to other bit-width models. The detailed comparison of *Double Rounding* with other multi-bit quantization methods is shown in Figure 1. Suppose the highest-bit and the current-bit are represented by $b_h$ and $b_c$ respectively. $\Delta_c = b_h - b_c$ is the difference between them. Here is the specific formulation of our proposed *Double Rounding* method:

$$\widetilde{\boldsymbol{W}}_h = \text{clip}(\left\lfloor \frac{\boldsymbol{W} - \boldsymbol{z}_h}{\boldsymbol{s}_h} \right\rceil, -2^{b_h - 1}, 2^{b_h - 1} - 1), \quad and \tag{2}$$

$$\widetilde{\boldsymbol{W}}_c = \text{clip}(\left\lfloor \frac{\widetilde{\boldsymbol{W}}_h}{2^{\Delta_c}} \right\rceil, -2^{b_c - 1}, 2^{b_c - 1} - 1), \quad \widehat{\boldsymbol{W}}_c = \widetilde{\boldsymbol{W}}_c \times \boldsymbol{s}_h \times 2^{\Delta_c} + \boldsymbol{z}_h \tag{3}$$

where $\boldsymbol{s}_h \in \mathbb{R}$ and $\boldsymbol{z}_h \in \mathbb{Z}$ denote the highest-bit quantization *scale* and *zero-point*, respectively. $\widetilde{\boldsymbol{W}}_h$ and $\widetilde{\boldsymbol{W}}_c$ represent the quantized weights of the highest-bit and current-bit, respectively. In practical inference, the division and multiplication by $2^{\Delta_c}$ can be efficiently executed through shift operations, which are cost-effective and convenient. Note that in our proposed method, the model can also be preserved at full precision, if the hardware can meet the requirements of floating-point values during inference. And the $\boldsymbol{z}_h$ is 0 for the weight quantization in this paper.

Generally, unlike weight quantization, activation quantization is an online quantization process. Consequently, the corresponding *scale* and *zero-point* values for different bit-widths can be learned individually. The quantization process for activation (Bhalgat et al., 2020) can be formulated as follows:

$$\widetilde{\boldsymbol{A}}_c = \text{clip}(\left\lfloor \frac{\boldsymbol{A} - \boldsymbol{z}_c}{\boldsymbol{s}_c} \right\rceil, 0, 2^{b_c} - 1), \quad \widehat{\boldsymbol{A}}_c = \widetilde{\boldsymbol{A}}_c \times \boldsymbol{s}_c + \boldsymbol{z}_c \tag{4}$$

where $\boldsymbol{s}_c \in \mathbb{R}$ and $\boldsymbol{z}_c \in \mathbb{Z}$ represent the quantization *scale* and *zero-point* of current-bit activation, respectively. Note that $\boldsymbol{z}_c$ is 0 for the ReLU activation function. $\boldsymbol{A}$ and $\widetilde{\boldsymbol{A}}_c$ denote the floating activation and the quantized activation using the current-bit model, respectively. Additionally, gradient equations for training multi-bit quantization are represented as follows:

$$\frac{\partial \widehat{\boldsymbol{Y}}}{\partial \boldsymbol{s}_c} \simeq \begin{cases} \left\lfloor \frac{\boldsymbol{Y} - \boldsymbol{z}_c}{\boldsymbol{s}_c} \right\rceil - \frac{\boldsymbol{Y} - \boldsymbol{z}_c}{\boldsymbol{s}_c} & if \ n < \frac{\boldsymbol{Y} - \boldsymbol{z}_c}{\boldsymbol{s}_c} < p, \\ n \quad or \quad p & otherwise. \end{cases} \quad \frac{\partial \widehat{\boldsymbol{Y}}}{\partial \boldsymbol{z}_c} \simeq \begin{cases} 0 & if \ n < \frac{\boldsymbol{Y} - \boldsymbol{z}_c}{\boldsymbol{s}_c} < p, \\ 1 & otherwise. \end{cases} \tag{5}$$

where $n$ and $p$ denote the lower and upper bounds of the integer range $[N_{min}, N_{max}]$ targeted for quantizing the weights or activations, respectively. $\boldsymbol{Y}$ represents the weights or activations of the model, and $\widehat{\boldsymbol{Y}}$ represents the dequantized weights or activations of the model.

### 3.2 *Multi-LR* FOR DIFFERENT BIT-WIDTHS

While previous multi-bit quantization methods can handle simpler scenarios, once-joint optimization encounters significant challenges, primarily due to the competitive relationship between the highest and lowest bit-widths, as noted by Xu et al. (2022). As each bit-width model reaches its optimal state, they simultaneously influence each other during training. Tang et al. (2022) also highlights substantial differences in convergence rates between the highest and lowest bit-widths in once-joint training of multi-bit models. We experimentally identify the root cause of this competitive relationship as the inconsistent updating of gradients between high-bit and low-bit quantization during training. We further observe significant discrimination in the gradient magnitudes of the activation's quantization scales across different bit-width models. Figure 2 gives an example of ResNet20 model on CIFAR-10 datasets. Notably, significant differences are evident across

different bit-widths, with an order of magnitude disparity observed between 2-bit and higher-bit representations. Additional statistical results for other networks can be found in Section C of the Appendices.

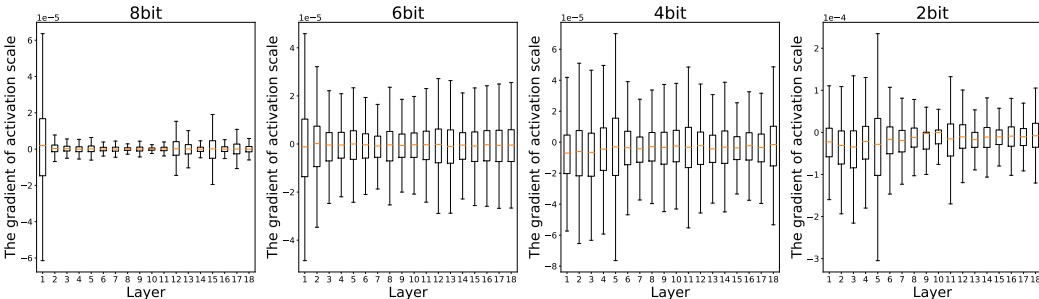

Figure 2: The scale gradient statistics of activation of ResNet20 model on CIFAR-10 dataset. Note that the outliers have been removed for clarity, and the values on the ordinate are represented on an exponential scale. The first and last layers of the model remain unquantized.

Motivated by the above observations, we introduce a novel training strategy for multi-bit models called *Multi-LR*. This strategy employs distinct learning rates for different bit-width models. Specifically, if the learning rate for the highest bit-width is $\lambda$, we set the learning rate for current lower bit-widths to $\lambda_c = \Gamma_c(\lambda)$, where $\Gamma_c(\cdot)$ represents a collapse function. The collapse function serves as a learning rate adjustment mechanism that maps higher-bit learning rates to lower-bit learning rates through a linear or nonlinear function. Through experimentation, we have found that adaptive *proportional scaling*, which depends on the difference between higher-bit and lower-bit values, is an optimal choice. For instance, if the learning rate for 8-bit is 0.01, it becomes 0.001 for 6-bit and 0.0001 for 4-bit. The scaling factor is typically set as a hyperparameter and commonly takes values as 0.1. As a result, this approach maintains the consistency of updating overall parameters with different bit-widths, leading to a more stable convergence during training and achieving optimal performance, as illustrated in Figure 3. Finally, when combining this strategy with the *cosine* learning schedule, the initial learning rates for different bit-width models exhibit varying magnitudes, which then gradually converge during the training process. For a detailed comparison between the previous multi-bit training approach and the *Multi-LR* approach, please refer to Algorithm 1 and Algorithm 2 in Section B of the Appendices.

### 3.3 MIXED-PRECISION SUPER-NET

Compared to multi-bit quantization models, mixed-precision super-net provide finer-grained control by adaptive configuring the bit-width at every layer of the model. For a detailed comparison of the training processes between these methods, please refer to Figure 5 in Section A of the Appendices. However, previous methods often involve substantial search costs to select the optimal subnetwork, typically using techniques like the greedy algorithm (Bulat & Tzimiropoulos, 2021) or genetic algorithm (Xu et al., 2022). To address this, we introduce a strategy for training the mixed-precision super-net using *weighted probability*, allowing for the rapid selection of the optimal subnetwork during inference. In contrast to conventional mixed-precision quantization approaches, which first determine the bit-widths configuration and then train the corresponding subnetwork, the mixed-precision super-net directly provides a suitable quantized subnetwork based on FLOPs or model size requirements without retraining or fine-tuning.

It may be unreasonable like Bulat & Tzimiropoulos (2021) to treat every layer equally for random bit-width selection due to the fact that the sensitivity (Dong et al., 2020) of each layer of the model is different. Consequently, it becomes essential to discriminate the selection frequency of different bit-widths for different layers. To maintain consistency between the training and decision-making processes, we exclusively consider sensitivity as a prior factor for bit allocation. It is important to emphasize that our approach does not incorporate any knowledge from the test dataset into the training process, ensuring that the model's generalization capability is preserved. Specifically, the Hessian trace is utilized to measure the sensitivity of each layer, and the Integer Linear

Programming (ILP) algorithm (Yao et al., 2021) is employed to find the optimal configurations. This allows us to select the optimal subnetwork without requiring additional training or fine-tuning. All the candidate configurations collectively constitute the Pareto Frontier optimal solutions for our method. Unlike Dong et al. (2020), our method can rapidly generate multiple candidate configurations for a given constraint by adapting the ILP algorithm after training the super-net. For an in-depth explanation of the mixed-precision training process, please consult Algorithm 3 in Section B of the Appendices, and the process for selecting the optimal subnetwork is outlined in Algorithm 4 in the same section. Experimental results confirm the effectiveness of the training strategy for the mixed-precision super-net with *weighted probability*, as discussed in Section 4.3.2.

Table 1: Accuracy comparisons of multi-bit on ImageNet datasets: including AdaBits, Any-Precision, Bit-Mixer, and MultiQuant. 'KD' denotes knowledge distillation, * denotes that this bit-width is not included in the training, – denotes the value is not published. The best result is shown in boldface.

| Model | Method | KD | Storage | Epoch | w8a8 | w6a6 | w4a4 | w3a3 | w2a2 | Float |
|-------|--------|-----|---------|-------|------|------|------|------|------|-------|
| ResNet18 | Any-Precision | ✓ | 32bit | 80 | 68.04 | * | 67.96 | * | 64.19 | 69.27 |
| | MultiQuant | ✓ | 32bit | 90 | 70.28 | 70.14 | 69.80 | 67.53 | 66.56 | 69.76 |
| | ours [8,6,4,2] | | 8bit | 90 | 70.79 | 70.82 | 70.43 | * | 66.65 | 70.41 |
| | ours [8,6,4,2] | ✓ | 8bit | 90 | **71.81** | **71.05** | **70.52** | * | **66.84** | 70.41 |
| | Bit-Mixer | ✓ | 4bit | 160 | * | * | 69.4 | 68.7 | 65.6 | 69.6 |
| | ours [4,3,2] | | 4bit | 90 | * | * | 69.73 | 69.20 | 66.20 | 70.41 |
| | ours [4,3,2] | ✓ | 4bit | 90 | * | * | **70.14** | **69.83** | **66.72** | 70.41 |
| ResNet50 | Any-Precision | | 32bit | 80 | 74.68 | * | 74.43 | * | 72.88 | 75.95 |
| | Any-Precision | ✓ | 32bit | 80 | 74.91 | * | 74.75 | * | 73.24 | 75.95 |
| | MultiQuant | ✓ | 32bit | 90 | 76.94 | 76.85 | 76.46 | 75.35 | 73.76 | 76.13 |
| | ours [8,6,4,2] | | 8bit | 90 | 76.88 | 76.67 | 75.74 | * | 72.38 | 76.7 |
| | ours [8,6,4,2] | ✓ | 8bit | 90 | **77.22** | **76.96** | 76.62 | * | **73.96** | 76.7 |
| | AdaBits | | 32bit | 150 | * | * | 76.1 | 75.8 | 73.2 | 75.9 |
| | Bit-Mixer | ✓ | 4bit | 160 | * | * | 75.2 | 74.9 | 72.7 | – |
| | ours [4,3,2] | | 4bit | 90 | * | * | 76.06 | 75.61 | 72.93 | 76.7 |
| | ours [4,3,2] | ✓ | 4bit | 90 | * | * | **76.68** | **75.96** | **73.52** | 76.7 |
| MobileNetV2 | AdaBits | | 8bit | 150 | 72.3 | 72.3 | 70.3 | * | * | 71.8 |
| | MultiQuant | ✓ | 32bit | 90 | 72.33 | 72.09 | 70.59 | * | * | 71.88 |
| | ours [8,6,4] | | 8bit | 90 | 72.42 | 72.06 | 69.92 | * | * | 71.14 |
| | ours [8,6,4] | ✓ | 8bit | 90 | **72.55** | **72.41** | **70.86** | * | * | 71.14 |
| | ours [8,6,4,2] | | 8bit | 90 | 70.98 | 70.70 | 68.77 | * | 50.43 | 71.14 |
| | ours [8,6,4,2] | ✓ | 8bit | 90 | 71.35 | 71.20 | 69.85 | * | **53.06** | 71.14 |

## 4 EXPERIMENTS

### 4.1 EXPERIMENTAL SETTINGS

In our experiments, we primarily focus on classification task using the ImageNet dataset (Deng et al., 2009). This choice is motivated by the fact that previous works have predominantly concentrated on classification tasks, including notable methods like AdaBits (Jin et al., 2020), Any-Precision (Yu et al., 2021), Bit-Mixer (Bulat & Tzimiropoulos, 2021), and MultiQuant (Xu et al., 2022). To enable meaningful comparisons, we conducted extensive experiments on both classical networks (ResNet18/50 (He et al., 2015)) and lightweight networks (MobileNetV2 (Sandler et al., 2018)). Our experiments cover once-joint quantization training for multi-bit and fine-grained mixed precision. We explore two candidate bit configurations, namely [8, 6, 4, 2] and [4, 3, 2]. In these configurations, each number represents the quantization level of the weight and activation layers. These choices are made to align with previous works, thus facilitating a better comparison, while also minimizing the training overhead. Like previous methods, we exclude batch normalization layers from quantization, and the first and last layers are kept at full precision.

We initialize the model with a floating-point pre-trained model for multi-bit joint training, and for mixed-precision super-net joint training, we initialize the model using the trained multi-bit model. All models are trained for 90 epochs using a cosine scheduler without a warm-up phase. We employ the *Adam* optimizer (Kingma & Ba, 2014) with a batch size of 256. The initial learning rate is set to $5e - 4$, and the weight decay is set to $5e - 5$. Data augmentation adheres to the standard set of transformations used for ImageNet dataset. This includes random cropping, resizing to $224 \times 224$ pixels, and random flipping. During evaluation, images are resized to $256 \times 256$ pixels and then center-cropped to a resolution of $224 \times 224$ pixels. For more specific details regarding the training process, please refer to Algorithm 2 and Algorithm 3 in Section B of the Appendices.

Table 2: Accuracy comparisons of mixed-precision on ImageNet dataset. 'KD' denotes knowledge distillation, * denotes that this bit-width is not included in the training, – denotes the value is not published.

| Model | Method | KD | Storage | Epoch | 8w8a | 6w6a | 4w4a | 3w3a | 2w2a | Float |
|---|---|---|---|---|---|---|---|---|---|---|
| ResNet18 | ours [8,6,4,2] | | 8bit | 90 | 70.38 | 70.31 | 70.01 | * | 65.50 | 70.41 |
| | Bit-Mixer | | 4bit | 160 | * | * | 69.0 | 68.4 | 64.0 | 69.6 |
| | Bit-Mixer | ✓ | 4bit | 160 | * | * | 69.2 | 68.6 | 64.4 | 69.6 |
| | ours [4,3,2] | | 4bit | 90 | * | * | 69.80 | 68.63 | 64.88 | 70.41 |
| | ours [4,3,2] | ✓ | 4bit | 90 | * | * | **70.03** | **69.32** | **65.17** | 70.41 |
| ResNet50 | ours [8,6,4,2] | | 8bit | 90 | 75.76 | 75.83 | 75.31 | * | 72.43 | 76.7 |
| | Bit-Mixer | ✓ | 4bit | 160 | * | * | 75.2 | 74.8 | 72.1 | – |
| | ours [4,3,2] | | 4bit | 90 | * | * | 75.64 | 75.36 | 72.47 | 76.7 |
| | ours [4,3,2] | ✓ | 4bit | 90 | * | * | **75.85** | **75.42** | **72.81** | 76.7 |

## 4.2 EXPERIMENTAL RESULTS

In this section, we primarily conduct a comparison between our proposed method and previous approaches on ImageNet dataset. The experimental results for multi-bit joint quantization are presented in Table 1, and the results for mixed precision super-net quantization are shown in Table 2. It's worth noting that the accuracy values of the FP32 models in the two tables aren't uniform due to the absence of open-source code or pre-trained models from previous works. To enable a standardized comparison, diligent efforts have been made to align the FP32 model's performance with the latest SOTA to ensure a meaningful comparison.

**Multi-Bit Joint Quantization**: Table 1 provides clear evidence that our proposed *double-rounding* multi-bit joint quantization method surpasses the previous SOTA methods on the ImageNet dataset. It's also worth noting that when considering candidate bit-widths, including 2-bit, the previous multi-bit joint quantization methods frequently experience convergence issues, particularly on MobileNetV2 during training. In contrast, our method exhibits the ability to converge, even with bit-widths such as [8,6,4,2]. We analyze that the superior performance of our method compared to previous methods may be attributed to the incorporation of two *rounding* operations. This approach can represent the full quantization values and helps reduce loss when transitioning between different bit-widths. Additionally, the *Multi-LR* training strategy we have introduced can mitigate the competitive relation between highest-bit and lowest-bit representations, as demonstrated in Figure 3. This leads to more stable convergence during multi-bit joint quantization training, and performance can be further enhanced through knowledge distillation (Kim et al., 2021).

**Mixed-Precision Super-Net Quantization**: The results of mixed-precision super-net are presented in Table 2. Similar to the multi-bit quantization experiments, our proposed mixed precision super-net with *Weighted Probability* joint quantization method also outperforms the previous SOTA methods on the ImageNet dataset. Since previous methods neither explore candidate bit-widths like [8,6,4,2] nor conduct experiments on the MobileNetV2 network, we mainly focus on discussing the mixed-precision super-net quantization of candidate bit-widths [4,3,2] on ResNet18/50. As a result, it appears to be effective in mixed-precision super-net joint training to consider sensitivity when randomly allocating bit-widths to different layers. The performance of mixed precision can also

be further enhanced through knowledge distillation (Kim et al., 2021). Additionally, we have also implemented the candidate bit-widths [8,6,4,2] for mixed-precision super-net experiments to serve as a valuable reference for future research in the community.

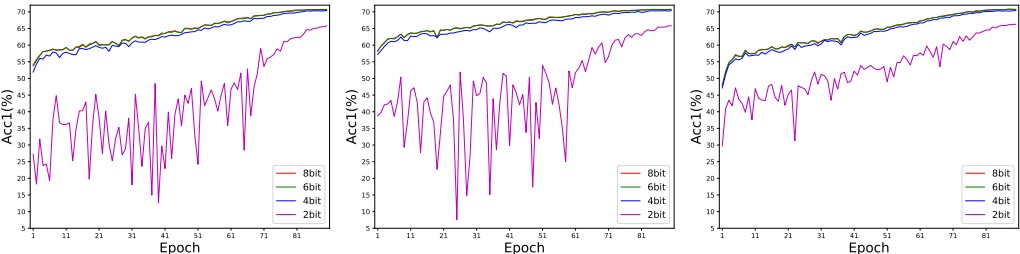

Figure 3: Comparison of training processes of different strategies of ResNet18 on ImageNet dataset:(from left to right) use 'shared scale of different bit-widths for activation', 'individual scales of different bit-widths for activation', and '*Multi-LR*'.

### 4.3 ABLATION STUDIES

#### 4.3.1 MULTI-BIT QUANTIZATION

In order to verify the effectiveness of our proposed *Multi-LR* training strategy, we conduct an ablation experiment using ResNet18 on the ImageNet dataset. As shown in Figure 3, applying the *Multi-LR* training strategy for multi-bit joint training clearly reduces the difference between convergence of the higher-bits and lowest-bit, resulting in a more stable training trend. Furthermore, it indicates that employing individual scales for activation allows for better alignment with the pre-trained model during the early stages of training, facilitating more comprehensive model convergence. We also conduct similar ablation experiments on other models and observe overall performance improvements, particularly for the 2-bit scenario. Detailed results are available in Table 3. It is obvious that the impact of using individual scales for different bit widths of activation on the final performance, and it also demonstrates the effectiveness of our proposed *Multi-LR* strategy in multi-bit joint training. Additionally, we have documented statistical results regarding the time required for multi-bit joint training compared to single-bit training, as shown in Table 4.

Table 3: Ablation studies of multi-bit joint training, among that ResNet20 is on CIFAR-10 dataset and other models are on ImageNet dataset. Ind. scale denotes using individual learnable scales for different bit widths about activation. *Multi-LR* denotes using '*Multi-LR*' training strategy.

| Model | Ind. scale | *Multi-LR* | [8,6,4,2] | | | | [4,3,2] | | | Float |
|---|---|---|---|---|---|---|---|---|---|---|
| | | | w8a8 | w6a6 | w4a4 | w2a2 | w4a4 | w3a3 | w2a2 | |
| ResNet20 | ✓ | | 92.01 | 92.07 | 91.92 | 88.57 | 91.74 | 91.19 | 87.51 | 92.36 |
| | ✓ | | 92.17 | 92.20 | 92.17 | 89.67 | 91.19 | 90.98 | 88.62 | 92.36 |
| | ✓ | ✓ | 92.25 | 92.32 | 92.09 | 90.19 | 91.79 | 91.83 | 88.88 | 92.36 |
| ResNet18 | ✓ | | 70.74 | 70.72 | 70.44 | 65.78 | 68.76 | 68.13 | 64.67 | 70.41 |
| | ✓ | | 70.75 | 70.71 | 70.34 | 65.83 | 69.38 | 68.74 | 65.62 | 70.41 |
| | ✓ | ✓ | 70.79 | 70.82 | 70.43 | 66.65 | 69.73 | 69.20 | 66.20 | 70.41 |
| ResNet50 | ✓ | | 76.06 | 75.93 | 75.50 | 70.70 | 74.72 | 74.72 | 68.90 | 76.7 |
| | ✓ | | 76.18 | 76.08 | 75.64 | 71.31 | 75.48 | 74.85 | 71.64 | 76.7 |
| | ✓ | ✓ | 76.88 | 76.67 | 75.74 | 72.38 | 76.06 | 75.61 | 72.93 | 76.7 |
| | | | [8,6,4,2] | | | | [8,6,4] | | | |
| MobileNetV2 | ✓ | | 70.05 | 70.85 | 68.08 | 45.00 | 72.06 | 71.87 | 69.86 | 71.14 |
| | ✓ | ✓ | 70.98 | 70.70 | 68.77 | 50.43 | 72.42 | 72.06 | 69.92 | 71.14 |

While previous findings suggested a reduction in multi-bit joint training time compared with single-bit training repeated of around $10\%$ (Du et al., 2020), we observed a more significant decrease, ranging from approximately $15\%$ to $20\%$. This conclusion once again reaffirms that multi-bit joint training demands less training cost, thereby conserving energy and reducing emissions.

Table 4: Ablation studies of comparing the cost of multi-bit joint training with separate bit training, averaged over three runs. 'N' denotes the number of separate bits.

| Model | Dataset | Bit-widths | Number of V100 | Epochs | Batchsize | GPU hours |
|-------|---------|------------|----------------|--------|-----------|-----------|
| ResNet18 | ImageNet | [8,6,4,2] | 4 | 90 | 256 | 65.3 |
| | | [4,3,2] | 4 | 90 | 256 | 45.7 |
| | | separate bits | 4 | 90 | 256 | $19.0 \times N$ |
| ResNet50 | ImageNet | [8,6,4,2] | 4 | 90 | 256 | 163.2 |
| | | [4,3,2] | 4 | 90 | 256 | 122.3 |
| | | separate bits | 4 | 90 | 256 | $51.6 \times N$ |

### 4.3.2 MIXED-PRECISION QUANTIZATION

To further verify the effectiveness of our proposed *weighted probability* strategy, we conduct an ablation experiment using ResNet18 on ImageNet dataset. The results can be observed from the right sub-figure in Figure 4, where the use of weighted probability demonstrates a significant improvement compared to not using weighted probability in the inference stage. Additionally, the overall Pareto optimal frontier of using weighted probability is higher, bringing us closer to the optimal solution.

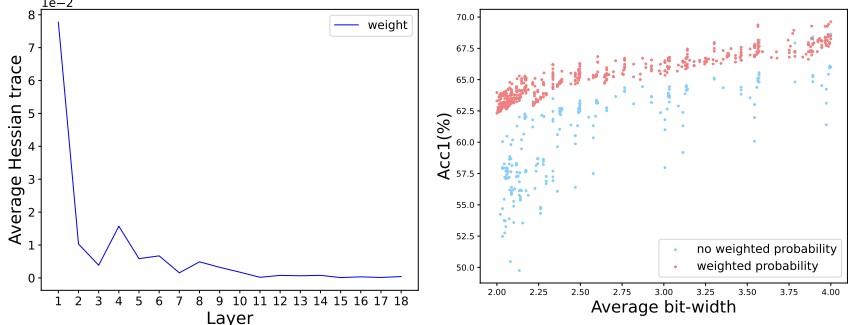

Figure 4: Left: Average Hessian trace of different layers in ResNet18. Right: Top-1 accuracy of mixed-precision given average bit-width of ResNet18 on ImageNet dataset.

## 5 CONCLUSION

In this paper, a novel scheme is presented for addressing the challenges of multi-bit joint training and mixed-precision super-net training. The approach aims to achieve an efficient and effective model compression technique to flexible adaptive different storage and computation requirements. Firstly, a new quantization method called *Double Rounding* is proposed for multi-bit joint training, such that only the highest-bit integer model needs to be stored. Secondly, by setting different learning rates for the varying bit-widths, the performance of multi-bit training is improved. Thirdly, the proposed weighted probability training strategy for mixed-precision super-net training enhances the generality of the proposed method. Furthermore, a decision-making method based on integer linear programming is developed to determine the optimal bit-width combination for different layers of the model, achieving approximate Pareto Frontier optimal solutions. Taken together, these strategies present a practical solution for flexible model compression, allowing deployment on diverse hardware platforms while preserving high performance.

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
