## A  COMPARISON OF DIFFERENT QUANTIZATION TYPES

To distinguish between traditional independent bit quantization, multi-bit joint quantization, and the optimization of mixed-precision super-net, we have visualized the detailed training process diagrams of different quantization types, as depicted in Figure 5. It is evident that for individual bit quantization when different bit configurations are considered, the complete quantization process needs to be repeated. Quantization-Aware Training (QAT) requires retraining, while Post-Training Quantization (PTQ) necessitates fine-tuning (Jacob et al., 2017). Such costs are unacceptable for the same model.

However, the multi-bit quantization method of once-joint training effectively addresses the problem of repeated training for individual bits, allowing for model adaptation, especially in the domain of autonomous driving, to respond to different scene changes. This enables a trade-off between model size, accuracy, and latency, reducing hardware power consumption and contributing to energy efficiency and emission reduction. Compared to once-joint multi-bit quantization, the mixed-precision super-net offers a more finely-grained adjustment of bit-width allocation for different layers, thus enhancing the overall model performance.

Additionally, the proposed *Double Rounding* quantization method in this paper facilitates the seamless switching of bits in different layers of mixed-precision super-net during inference to achieve a more reasonable trade-off between model size or FLOPs and accuracy. Importantly, the entire bit-switching process requires no retraining or fine-tuning.

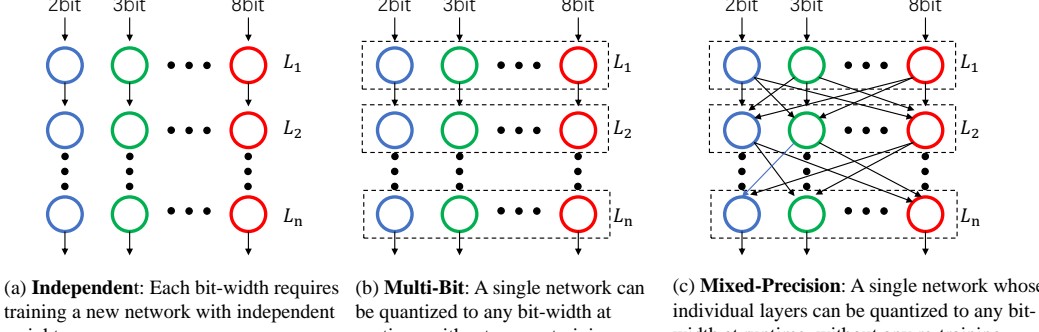

(a) **Independent**: Each bit-width requires training a new network with independent weights.

(b) **Multi-Bit**: A single network can be quantized to any bit-width at runtime, without any re-training.

(c) **Mixed-Precision**: A single network whose individual layers can be quantized to any bit-width at runtime, without any re-training.

Figure 5: Comparison between different network quantization types

## B  ALGORITHMS

The algorithm implementations of multi-bit joint quantization and mixed-precision super-net optimization are provided in this section. Firstly, we discuss the differences between the conventional multi-bit joint training method and the multi-bit joint training method based on *Multi-LR* proposed in this paper. Next, we elaborate on the training implementation details of the mixed-precision super-net with weighted probability. Lastly, we implement a decision-making scheme to efficiently select the optimal subnetwork based on the trained mixed-precision super-net.

### B.1  COMPARISON BETWEEN THE CONVENTIONAL MULTI-BIT TRAINING APPROACH AND THE *Multi-LR* APPROACH

In general, during multi-bit joint quantization training, all candidate models with different quantization precisions share the same model weights, typically the highest-bit model.

Since it can be difficult to achieve convergence when losses calculated by different bit-widths directly update the shared weights separately, previous conventional multi-bit joint quantization training primarily acquired the corresponding loss by forwarding each round of mini-batch data multiple times. The model size for each forward pass is determined based on the current bit-width.

In other words, during each round of training iterations, the model goes through the same number of iterations as the candidate bit-widths, accumulating a total loss, and then updating the parameters accordingly. For specific implementation details, please refer to Algorithm 1.

We have discovered that the reason for the lack of convergence when updating shared parameters according to the corresponding bit-width in each round is the inconsistency in the update pace of different bit-widths. In other words, the gradient magnitudes of quantization scales of different bit-widths vary, making it hard to attain stable convergence during training.

To address this issue, we introduce an adaptive approach to alter the learning rate for different bit-widths, aiming to achieve a consistent overall update pace. This modification allows us to directly update the shared parameters after calculating the loss for the current bit at a time. We have named this training strategy *Multi-LR*, and it has been experimentally proven to result in improved performance for multi-bit models.

In the actual implementation, we update both the parameters of weights and the parameters of the quantization scales simultaneously using dual optimizers. We also set the weight-decay of the quantization scales to 0 to achieve more stable convergence. This is done because the shared scales learned during multi-bit joint training are more sensitive (Xu et al., 2022) than those in individual-bit quantization training. For specific implementation details, please refer to Algorithm 2.

---

**Algorithm 1** Conventional training approach

**Require:** Candidate bit-widths set $b_{c} \in \mathbb{B}$
1: Initialize: Model $M$ with floating-value, the quantization scales $S$ including $s_{h}$ of weights and $s_{c}$ of activations, BatchNorm layers: $\{BN\}_{c=1}^{n}$, optimizer: $optim(\boldsymbol{W}, S)$;
2: For one epoch;
3: Sample mini-batch data $(\mathbf{x}, \mathbf{y}) \in \{D_{train}\}$
4: **for** $b_{c}$ in $\mathbb{B}$ **do**
5:   **for** each quantization layer **do**
6:     $\widehat{w}_{c} = dequant(quant(w, s_{h}))$
7:     $\widehat{x}_{c} = dequant(quant(x, s_{c}))$
8:     $o_{c} = M(\widehat{w}_{c}, \widehat{x}_{c})$
9:   **end for**
10:   Update $BN_{c}$ layers
11:   Compute loss: $\mathcal{L}_{c} = CE(\mathbf{O_{c}}, \mathbf{y})$
12:   Compute gradients: $\mathcal{L}_{c}.backward()$
13: **end for**
14: Update weight and scales using accumulated gradients: $optim.step()$
15: Clear gradient: $optim.zero\_grad()$;

---

**Algorithm 2** *Multi-LR* training approach

**Require:** Candidate bit-widths set $b_{c} \in \mathbb{B}$
1: Initialize: Model $M$ with floating-value, the quantization scales $S$ including $s_{h}$ of weights and $s_{c}$ of activations, BatchNorm layers: $\{BN\}_{c=1}^{n}$, optimizers: $optim_{1}(\boldsymbol{W})$, $optim_{2}(S, wd = 0)$, learning rate: $\lambda$, the collapse function: $\Gamma_{c}(\cdot)$;
2: For one epoch;
3: Sample mini-batch data $(\mathbf{x}, \mathbf{y}) \in \{D_{train}\}$
4: **for** $b_{c}$ in $\mathbb{B}$ **do**
5:   **for** each quantization layer **do**
6:     $\widehat{w}_{c} = dequant(quant(w, s_{h}))$
7:     $\widehat{x}_{c} = dequant(quant(x, s_{c}))$
8:     $o_{c} = M(\widehat{w}_{c}, \widehat{x}_{c})$
9:   **end for**
10:   Update $BN_{c}$ layers
11:   Compute loss: $\mathcal{L}_{c} = CE(\mathbf{O_{c}}, \mathbf{y})$
12:   Compute gradients: $\mathcal{L}_{c}.backward()$
13:   Compute learning rate: $\lambda_{c} = \Gamma_{c}(\lambda)$
14:   Update weights and scales:
    $optim_{1}.step(\lambda)$
    $optim_{2}.step(\lambda_{c})$
15:   Clear gradient:
    $optim_{1}.zero\_grad()$
    $optim_{2}.zero\_grad()$
16: **end for**

---

**Note:** $n$ represents the number of candidate bit-widths, $CE$ denotes $CrossEntropyLoss()$, $wd$ denotes *weight decay*, $\boldsymbol{W}$ denotes weights of the model.

## B.2 EFFICIENT TRAINING OF MIXED PRECISION

Unlike multi-bit joint quantization, the training of the mixed-precision super-net is more random. In multi-bit training, the bit-widths calculated in each round are fixed, such as [8,6,4,2]. However, in mixed-precision super-net training, the bit-widths updated in each round are not fixed, for example,

[8,random-bit,2], similar to the *sandwich* strategy of Yu et al. (2018). Therefore, mixed precision training often requires more training epochs to reach convergence compared to multi-bit training.

Bit-mixer (Bulat & Tzimiropoulos, 2021) conducts the same probability of randomly selecting bit-width for different layers. However, we take the sensitivity of each layer into consideration which uses sensitivity (*e.g.* Hessian's trace (Dong et al., 2020)) as a weight to measure the random selection probability of different layers. For more sensitive layers, preference is given to higher-bit widths, and vice versa. We refer to this training strategy as the *weighted probability* approach for optimizing the mixed-precision super-net. Specific implementation details can be found in Algorithm 3. It's worth noting that we also consider the sensitivities of different layers when selecting the optimal subnetwork to ensure consistent decisions after training the mixed-precision super-net.

In additionally, different from multi-bit joint training, the BN layers are replaced by TBN (Transitional Batch-Norm) (Bulat & Tzimiropoulos, 2021), which compensates for the distribution shift between adjacent layers that are quantized to different bit-widths. The main difference between our proposed method and Bit-Mixer lies in the 12th line of Algorithm 3. The average trace of the Hessian matrix of each layer is considered when switching bits, which makes the training process of mixed precision consistent with the decision-making for selecting a sub-network by Integer Linear Programming (ILP) algorithm (Yao et al., 2021).

---

**Algorithm 3** Mixed-precision training approach

---

**Require:** Candidate bit-widths set $b_c \in \mathbb{B}$, the average hessian trace of different layers of float-point model: $t_l \in \{T\}_{l=1}^{L}$

1: Initialize: Model $M$ with floating-value, the quantization scales $S$ including $s_h$ of weights and $s_c$ of activations, BatchNorm layers: $\{BN\}_{c=1}^{n^2}$, optimizer: $optim(\boldsymbol{W}, S)$, the probability of bit-switching: $\sigma_p = 0$, the probability threshold of bit-switching: $\sigma_k = 3/4$, the total number of epochs: $e_p$, the epoch of unincreasing probability of bit-switching: $e_p = 2/3 \cdot e_{total}$, the probability step of bit-switching: $\delta_p = \sigma_k/e_p$, the current number of epochs: $e_{cur}$;

2: For one epoch;
3: Sample mini-batch data $(\mathbf{x}, \mathbf{y}) \in \{D_{train}\}$
4: **for** $b_c$ in $\mathbb{B}$ **do**
5:   **if** $\sigma_p >= \sigma_k$ **then**
6:     $\sigma_p = \sigma_k$
7:   **else**
8:     $\sigma_p = \delta_p \cdot e_{cur}$
9:   **end if**
10:   **for** each quantization layer **do**
11:     **if** $random() >= (1 - \sigma_p)$ **then**
12:       $b_c = random.choice(\mathbb{B}, \boldsymbol{t}_l)$
13:     **end if**
14:     $\widehat{w}_c = dequant(quant(w, \boldsymbol{s}_h))$
15:     $\widehat{x}_c = dequant(quant(x, \boldsymbol{s}_c))$
16:     $o_c = M(\widehat{w}_c, \widehat{x}_c)$
17:   **end for**
18:   Update $BN_c$ layers
19:   Compute loss: $\mathcal{L}_c = CE(\mathbf{O_c}, \mathbf{y})$
20:   Compute gradients: $\mathcal{L}_c.backward()$
21: **end for**
22: Update parameters using accumulated gradients: $optim.step()$; $optim.zero\_grad()$

---

**Note:** $n$ represents the number of candidate bit-widths.

### B.3 DECISION-MAKING OF CANDIDATE BIT-WIDTHS OF MIXED PRECISION

After training the mixed-precision super-net, the next step is to select the appropriate optimal subnetwork based on criteria like model size or FLOPs for actual deployment and inference. Optimizing mixed-precision supernets is discussed in Section B.2, and it takes into account the sensitivities of different layers when considering the random selection of bit-widths.

To achieve solutions for bit allocation candidates under given FLOPs or model size conditions, we employ the Iterative Pareto Learning (IPL) approach. Since each IPL run can provide only one solution, we obtain multiple solutions by altering the types of different average bit-widths. This enables us to create several subnets and form a Pareto optimal frontier. From this frontier, we can select the appropriate subnet for deployment. For a detailed step-by-step process, please refer to Algorithm 4.

---

**Algorithm 4** Decision-Making of candidate bit-widths of mixed prccision

---

**Input:** Given candidates of average bit-width: $\omega_j \in \{\Omega\}_{j=1}^m$, candidate bit-widths set $b_c \in \mathbb{B}$,

the average hessain trace and parameter of different layers of float-point model:$t_l \in \{T\}_{l=1}^L$,

$n_l \in \{N\}_{l=1}^L$

**Output:** Candidate bit-widths for different layers:$\boldsymbol{p}_k \in \{P\}_{k=1}^{K \ll \mathbb{B}^L}$

1:  **for** $\omega_j$ in $\Omega$ **do**
2:      Objective function: $O_h = \sum_{l=1}^L \frac{t_l}{n_l} \cdot l_{b_c}$
3:      Constraints: $\omega_j == \frac{\sum_{l=1}^L l_{b_c}}{L}$
4:      The solve of problem: $\boldsymbol{s}_1 = pulp.solve()$ and $S.append(\boldsymbol{s}_1)$
5:      **for** $l_{b_c}$ in $p_1$ **do**
6:          **for** $b_c$ in $\mathbb{B}$ **do**
7:              **if** $b_c < max(s_1)$ and $b_c \neq l_{b_c}$ **then**
8:                  Add constraint: $l_{b_c} == b_c$
9:                  Problem solve: $\boldsymbol{p}_c = pulp.solve()$
10:                 **if** $\boldsymbol{s}_c$ not in $S$ **then**
11:                     $P.append(\boldsymbol{p}_c)$
12:                 **end if**
13:                 Pop last constraint
14:             **end if**
15:         **end for**
16:     **end for**
17: **end for**
18: **return** $P$

---

## C  THE GRADIENT STATISTICS OF LEARNABLE SCALE OF QUANTIZATION

In this section, we analyze the changes in gradients of the learnable scale for different models during the training process. Figure 6 and Figure 7 display the gradient statistical results for ResNet20 on CIFAR-10. Similarly, Figure 8 and Figure 9 show the gradient statistical results for ResNet18 on ImageNet, and Figure 10 and Figure 11 present the gradient statistical results for ResNet50 on ImageNet. These figures reveal a similarity in the range of gradient changes between higher-bit quantization and 2-bit quantization. Notably, they illustrate that the value range of 2-bit quantization is noticeably an order of magnitude higher than the value ranges of higher-bit quantization.

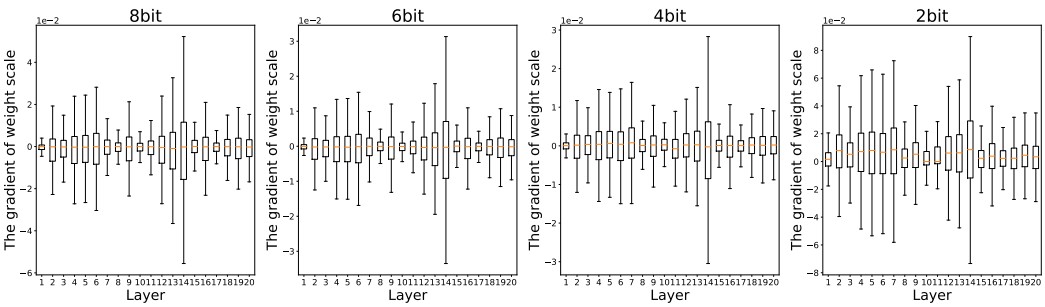

Figure 6: The scale gradient statistics of weight of ResNet20 on CIFAR-10 dataset. Note that the outliers are removed for exhibition.

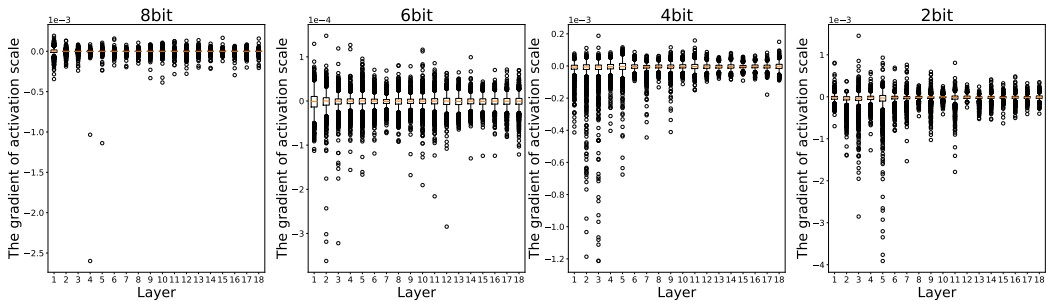

Figure 7: The scale gradient statistics of activation of ResNet20 on CIFAR-10 dataset. Note that the first and last layers are not quantized.

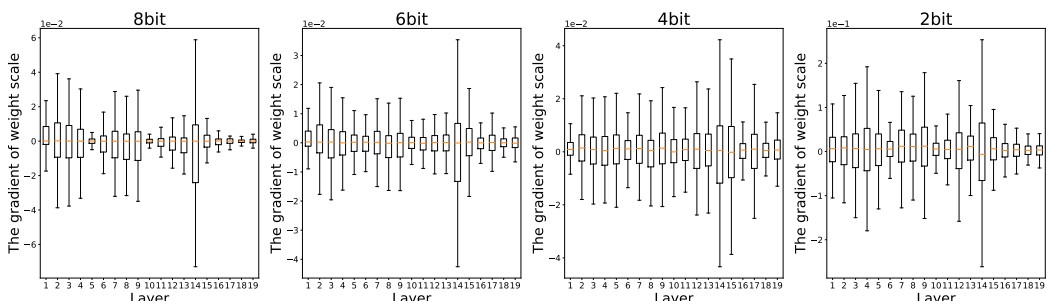

Figure 8: The scale gradient statistics of weight of ResNet18 on ImageNet dataset. Note that the outliers are removed for exhibition.

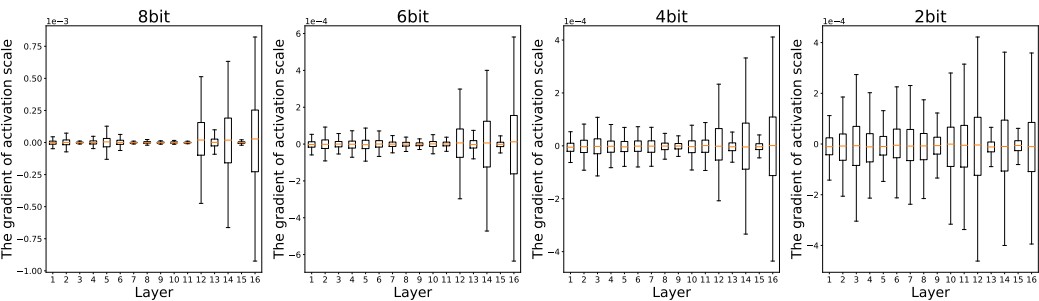

Figure 9: The scale gradient statistics of activation of ResNet18 on ImageNet dataset. Note that the outliers are removed for exhibition.

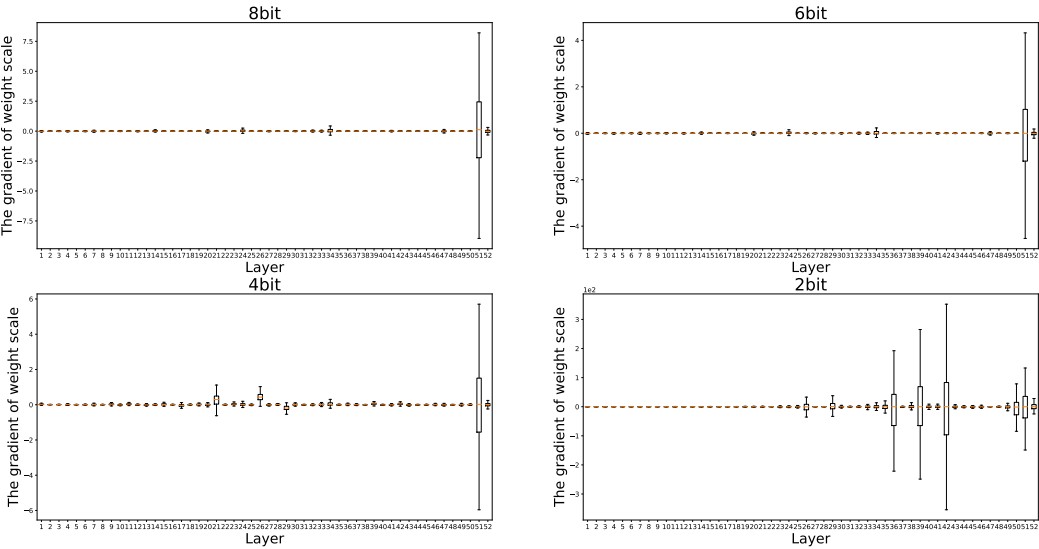

Figure 10: The scale gradient statistics of weight of ResNet50 on ImageNet dataset. Note that the outliers are removed for exhibition, and the first and last layers are not quantized.

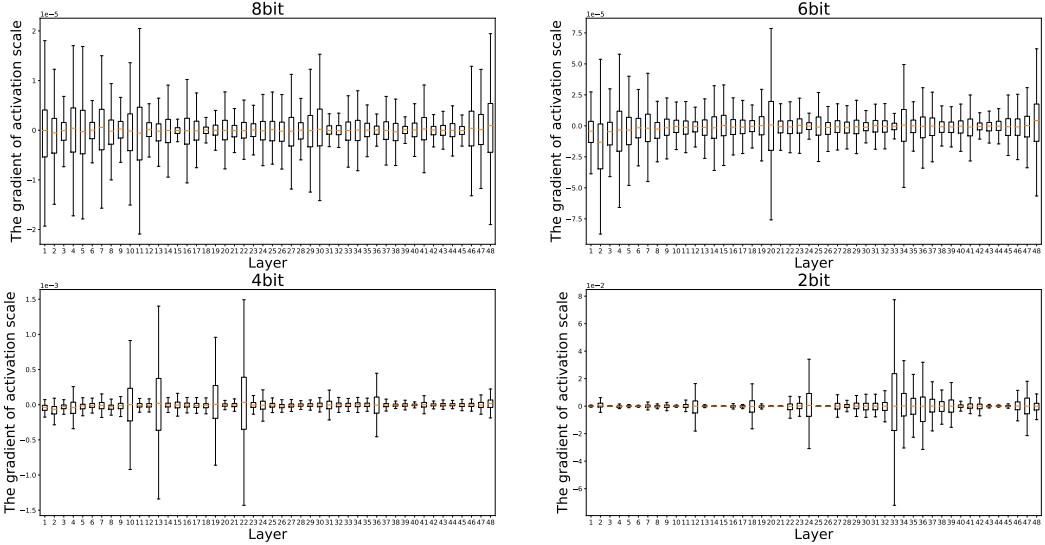

Figure 11: The scale gradient statistics of activation of ResNet50 on ImageNet dataset. Note that the outliers are removed for exhibition.