# OpenReview forum: "Double Rounding Quantization for Flexible Deep Neural Network Compression"
_ICLR.cc/2024/Conference — ICLR 2024 Conference Withdrawn Submission_

### Official Review · Reviewer_fWCS · 2023-10-25

**Soundness:** 3 good
**Presentation:** 3 good
**Contribution:** 2 fair
**Rating:** 5
**Confidence:** 5

**Summary:**

The paper introduces a new method for multi-bit joint training and mixed-precision super-net training in deep neural networks. This method aims to make model compression more adaptable for varying hardware and storage needs. The authors propose a Double Rounding quantization technique, which only requires storing the highest-bit integer model, making it more storage-efficient. They address the challenge of inconsistent gradients in multi-bit training by adaptively adjusting learning rates for different bit-widths. Additionally, a weighted probability training strategy for mixed-precision super-nets is introduced, improving the method's versatility. The paper also presents a decision-making approach using integer linear programming to find the best bit-width combination for different model layers, targeting optimal solutions. Experiments on CIFAR-10 and ImageNet demonstrate the effectiveness of the proposed method.

**Strengths:**

The paper is well written and easy to follow. The proposed double rounding quantization technique, which offers a storage-efficient solution by only necessitating the storage of the highest-bit integer model.

**Weaknesses:**

The paper has several notable weaknesses, including lack of empirical evidence, ambiguities in methodology, unclear contributors to performance, missing Baseline, etc. Please see questions for details.

**Questions:**

1.	In Section 3.2, the authors attribute the notable divergence in convergence rates between the highest and lowest bit-widths during once-joint training of multi-bit models to the inconsistency in gradient updates between high-bit and low-bit quantization phases. However, this assertion lacks empirical evidence, as no experimental results are presented to highlight these inconsistent gradients. Additionally, the introduced Multi-LR approach, which adjusts learning rates based on different bit-widths, is heuristic in nature. It would be beneficial to understand if there is an underlying rationale or guiding principle for the selection of these learning rates to ensure they are both effective and justifiable.

2.	In Section 3.3, the authors introduce the use of weighted probability for the supernet training. However, the methodology behind computing the sampling probability for various bit-widths within a given layer is not explicitly explained. This omission can lead to ambiguities in replicating the approach and understanding its full implications. Providing a clearer, step-by-step computation process would enhance the reproducibility of the proposed method.

3.	In Section 3.3, the authors mention the capability of their method to swiftly produce multiple candidate configurations under specified constraints by adapting the ILP (Integer Linear Programming) algorithm post super-net training. Yet, this assertion lacks empirical backing as the section doesn not offer any associated experimental results. Providing tangible evidence or case studies would substantiate this claim.

4.	The authors propose double rounding quantization that only keeps the highest bitwidth model instead of the full-precision counterpart. However, the performance drop brought from double rounding quantization is not clearly investigated. A deeper dive into the proposed method on the model's accuracy and efficiency would have provided a more comprehensive understanding of its real-world applicability and limitations.

5.	In Figure 2, the authors present the gradient statistics of activation scales for ResNet20, offering insights into the network's behavior. However, a critical aspect that is not clarified is the initialization of these scales. For a fair and meaningful comparison, it is essential to ascertain whether all scales started from the same initialization point.

6.	In Tables 1 and 2, the proposed method shows a significant performance improvement over the state-of-the-art methods. However, the specific components of the proposed method that primarily drive this performance enhancement remain ambiguous. A breakdown or ablation study highlighting the individual contributions of each component would provide deeper insight into the key drivers behind the observed improvements.

7.	The experimental section appears to lack a crucial baseline comparison. It would be valuable to understand how the proposed method stacks up against an independent approach that trains different bit-widths separately. Such a comparison would shed light on the relative efficacy and advantages of the proposed joint training technique.

8.	A pivotal detail seems to be overlooked in the paper. It is unclear how many samples the authors utilized to compute the Hessian trace across various layers.

---

> ### Author Response · Authors · 2023-11-13
>
> We would like to thank you for your careful reading, helpful comments, and constructive suggestions, which have significantly improved the presentation of our manuscript. We have carefully addressed all comments, and below are our responses:
>
> **Q1:** In Section 3.2, ... this assertion lacks empirical evidence, as no experimental results are presented to highlight these inconsistent gradients. Additionally, the introduced Multi-LR approach, which adjusts learning rates based on different bit-widths, is heuristic in nature. ...
>
> **A:** Thanks for your valuable feedback. We apologize for any confusion caused by the imprecise expression in Section 3.2. To clarify, our intended statement is that the gradient of the quantization scale between different bits during multi-bit training exhibits inconsistency, with the most noticeable difference between higher and lower bits, as illustrated in Figure 2. We will address this more accurately in the revision to enhance clarity. Regarding the Multi-LR method, we acknowledge its heuristic nature and recognize it as a parameter-tuning technology. We consider it a potential research direction for exploring adaptive adjustment strategies, in which the learning rates of different bits are learnable during training.
>
> **Q2:** In Section 3.3, ... the methodology behind computing the sampling probability for various bit-widths within a given layer is not explicitly explained. ...
>
> **A:** Thanks for your valuable observation. We acknowledge that due to space constraints, the specific methodology for computing sampling probabilities of various bit-widths within a given layer in supernet training is detailed in Algorithm 3 in the appendix. For a step-by-step calculation process, we invite you to refer to the 'W2' reply to reviewer mFBC for a comprehensive understanding.
>
> **Q3:** In Section 3.3, ... ILP (Integer Linear Programming) lacks empirical backing as the section does not offer any associated experimental results...
>
> **A:** Thanks for your valuable feedback. Specific details on the implementation can be found in Algorithm 4 in the appendix, and we recommend referring to 'W3' in the reply to reviewer mFBC. While the details may not be explicitly outlined in the current section, we are actively considering open sourcing our code. This will provide an opportunity for the community to replicate and validate our method's capabilities through practical experimentation.
>
> **Q4:** ... the performance drop brought by double rounding quantization is not clearly investigated...
>
> **A:** Thanks for your valuable perspective. It's crucial to clarify that our double rounding quantization method can save not only the highest bit-width model but also the full-precision model (32bit). Indeed, this approach often yields better results than saving only the highest bit-width model but needs larger storage.
>
> **Q5:** In Figure 2, ... a critical aspect that is not clarified is the initialization of these scales. For a fair and meaningful comparison, it is essential to ascertain whether all scales started from the same initialization point.
>
> **A:** Thanks for your comment. The initialization of the activated quantization scales for ResNet20 is consistent. All scales start from the same initialization point, utilizing the initialization operation of the LSQ[1] quantization method.
>
> **Q6:** In Tables 1 and 2, ...  A breakdown or ablation study highlighting the individual contributions of each component would provide deeper insight into the key drivers behind the observed improvements.
>
> **A:** Thanks for your insightful comment. In Table 1, focusing on multi-bit performance comparison, the observed performance improvements primarily stem from two key components: the newly proposed Double Rounding quantization method and the Multi-LR training strategy developed based on Double Rounding. For a detailed ablation study, please refer to Table 3 in this paper. Similarly, in Table 2, which centers on mixed precision performance, the enhancement can be attributed to the weighted probability training strategy based on Double Rounding. For a detailed ablation study, please refer to the right subfigure of Figure 4 in this paper.
>
> **Q7:** The experimental section appears to lack a crucial baseline comparison...
>
> **A:** Thanks for your valuable suggestion. Considering the results provided by the LSQ and LSQ+[2] papers, we omitted a separate training comparison in the current version. We will include results from separate training for different bit-widths in the revision.
>
> **Q8:** ...It is unclear how many samples the authors utilized to compute the Hessian trace across various layers.
>
> **A:** Thanks for your reminder. To compute the Hessian trace across various layers, we utilized 1000 training images.  Please refer to "W2" in our reply to reviewer mFBC.
>
> [1] Learned step size quantization
> [2] Lsq+: Improving low-bit quantization through learnable offsets and better initialization

---

> > ### Author Response · Authors · 2023-11-17
> >
> > Dear anonymous reviewer,
> >
> > Thank you for your constructive comments and valuable suggestions to improve this paper. Do my responses address your concerns? If you have any questions, please feel free to contact us. We're sorry to bother you again.
> >
> > Thank you very much.
> >
> > Best regards,
> >
> > Authors

---

### Official Review · Reviewer_DGQc · 2023-10-31

**Soundness:** 3 good
**Presentation:** 3 good
**Contribution:** 2 fair
**Rating:** 5
**Confidence:** 4

**Summary:**

The key innovations in the study are as follows: "Double Rounding" is proposed to maintain integer-weight parameter storage without compromising representation values. "Multi-LR" introduces a training strategy for multi-bit models that effectively reduces the training convergence gap between high-precision and low-precision models. "Weighted Probability" determines the access probability of bit-width for each layer based on the layer's sensitivity, aligning with the subnetwork's decision-making process during inference. Experimental results on ImageNet datasets demonstrate that the proposed method surpasses state-of-the-art techniques across various mainstream network architectures.

**Strengths:**

1. The paper exhibits a well-defined structure, making it easy to navigate and understand.

2. The primary objective is to address the complex issue of multi-bit quantization and demonstrate notable performance improvements compared to similar methods.

**Weaknesses:**

1. I find the distinction between Adabits and the proposed Double Rounding in the figure somewhat minimal. It seems that the primary difference lies in the altered value range. Could these two methods essentially be equated with one having a zero point and a different scale value?

2. It may be beneficial to include the algorithm for weight probability in the main paper. This approach could reduce the volume of explanatory text, ultimately enhancing clarity for readers.

3. Evaluating the proposed method on large transformer models as well as tiny models, that are particularly susceptible to the effects of quantization, would provide valuable insights.

4. In my overall assessment, I believe that the three proposed techniques may fall short of meeting the publication standards.

**Questions:**

1. Could you provide further clarification regarding the distinction between Adabits and Double Rounding?

2. The results presented in Table 1 raise the question of whether Knowledge Distillation (KD) plays a more significant role than the three proposed techniques.

3. Table 1 exclusively presents uniform bit-width results. Is there a specific reason for not including mixed precision results in the table?

4. In Table 4, the epoch duration remains consistent, but there is a variance in training cost. Can you clarify the specific factor or factors that account for this difference?

---

> ### Author Response · Authors · 2023-11-13
>
> We would like to thank you for your careful reading, helpful comments, and constructive suggestions, which have significantly improved the presentation of our manuscript. We are delighted with the identification of the novelty and effectiveness of the proposed method. We have carefully addressed all comments, and below are our responses:
>
> **W1:** I find the distinction between Adabits and the proposed Double Rounding in the figure somewhat minimal. It seems that the primary difference lies in the altered value range. Could these two methods essentially be equated with one having a zero point and a different scale value?
>
> **A:** Thanks for your observation. The distinction between Adabits and Double Rounding lies in the handling of scale values. As mentioned in our response to Reviewer GakW, Adabits determines the quantization scale based on the corresponding bit, while our Double Rounding allows for a learned, non-fixed scale. Furthermore, regarding the zero points, which are also learned. Because there is no ReLU function after the Projection layer in the MobileNetV2 block, we employ an asymmetric quantization method, drawing inspiration from the quantization strategy of LSQ+[1].
>
> **W2:** It may be beneficial to include the algorithm for weight probability in the main paper. This approach could reduce the volume of explanatory text, ultimately enhancing clarity for readers.
>
> **A:** Thanks for your valuable feedback. We will carefully reconsider the organization of the method section and work on a revised version that better accommodates your recommendation.
>
> **W3:** Evaluating the proposed method on large transformer models as well as tiny models, that are particularly susceptible to the effects of quantization, would provide valuable insights.
>
> **A:** Thank you for your insightful suggestion. Unfortunately, due to time constraints, we are unable to conduct experiments on large transformer models and tiny models as you suggested. We appreciate your understanding and recognize the potential value of exploring these aspects in future work.
>
> [1] Lsq+: Improving low-bit quantization through learnable offsets and better initialization
>
> **Q1:** Could you provide further clarification regarding the distinction between Adabits and Double Rounding?
>
> **A:** Please refer to our response to W1.
>
> **Q2:** The results presented in Table 1 raise the question of whether Knowledge Distillation (KD) plays a more significant role than the three proposed techniques.
>
> **A:** Thanks for your insightful observation. Knowledge Distillation (KD) is an additional enhancement performance technique in our method. While KD contributes to improved results, it alone cannot achieve the multi-bit joint training and ensure the lossless switching between different bits. Furthermore, ResNet18 in Table 1, demonstrates that our method without utilizing KD  surpasses the performance of other methods that rely on KD. To ensure a fair comparison, we also provide results combining our proposed techniques with KD technology. We hope this clarification addresses your concerns.
>
> **Q3:** Table 1 exclusively presents uniform bit-width results. Is there a specific reason for not including mixed precision results in the table?
>
> **A:** Thanks for your insightful question. The decision to present uniform bit-width results in Table 1 is rooted in the different training methods and application goals of multi-bit and mixed precision quantization. Multi-bit quantization focuses on bit switching for the overall model, whereas mixed precision quantization involves more fine-grained layer-wise bit switching. Directly comparing these two approaches would be unfair due to their inherent differences. Please refer to both Table 1 and Table 2, where you'll observe that the overall mixed precision results for the same model tend to be lower than those achieved with multi-bit quantization. This is because mixed precision requires optimizing a much larger space compared to multi-bit. Notably, many previous methods, including AdaBits and Any-Precision, primarily explore once-for-all multi-bit results without delving into mixed-precision results.
>
> **Q4:** In Table 4, the epoch duration remains consistent, but there is a variance in training cost. Can you clarify the specific factors that account for this difference?
>
> **A:** The variation in training cost observed in Table 4 is primarily influenced by the number of candidate bits. During training, each iteration involves forwarding the model a number of times equal to the candidate bits, followed by a single parameter update. So, the training cost tends to increase with a higher number of candidate bits. Because each iteration only needs to read batch size data once, the time cost of multi-bit training remains relatively lower compared to the cost of separate bit training performed multiple times. For a detailed understanding of the training process, please refer to Algorithm 1 or Algorithm 2 in the appendix.

---

> > ### Author Response · Authors · 2023-11-17
> >
> > Dear anonymous reviewer,
> >
> > Thank you for your constructive comments and valuable suggestions to improve this paper. Do my responses address your concerns? If you have any questions, please feel free to contact us. We're sorry to bother you again.
> >
> > Thank you very much.
> >
> > Best regards,
> >
> > Authors

---

### Official Review · Reviewer_GakW · 2023-11-01

**Soundness:** 3 good
**Presentation:** 3 good
**Contribution:** 2 fair
**Rating:** 5
**Confidence:** 5

**Summary:**

A multi-bit quantization framework (Double Quantization) is proposed, which quantizes a pre-trained model for once and enables inference with different pre-defined bit-width. To help convergence, a Multi-LR method is introduced to use seperate learning rate for each bit-width. Mixed-precision is also studied.

**Strengths:**

This paper deals with an important problem of network quantization, i.e., the multi-bit quantization problem. The proposed Multi-LR method seems to be useful for stable training. Experiments with various bit-widths and mix-precision results are provided.

**Weaknesses:**

The proposed multi-bit quantization framework consists of three main parts, i.e., the Double Rounding quantization scheme, the Multi-LR learning rate selection method, and the Weighted Probability mixed-precision method. However, these improvements seems to be a little bit incremental.
- I didn't see the difference between the double rounding quantization and adabit quantization. The adabit quantization can also represent with [−1, 1] but not limited to [0, 1].
- The multi-lr method is a hyper-parameter tuning, which is more like a tuning trick to me.
- Both mixed-precision quantization and mixed-precision based on multi-bit quantization have been widely studied in previous works.

The improvements over Adabit are not quite significant if KD is not used.

**Questions:**

Does all baseline methods use the same pre-trained model in Table-1? The full-precision baseline accuracy should be reported.

---

> ### Author Response · Authors · 2023-11-13
>
> We would like to thank you for your careful reading, helpful comments, and constructive suggestions, which have significantly improved the presentation of our manuscript. We have carefully addressed all comments, and below are our responses:
>
> **W1:** The proposed multi-bit quantization framework consists of three main parts, i.e., the Double Rounding quantization scheme, the Multi-LR learning rate selection method, and the Weighted Probability mixed-precision method. However, these improvements seems to be a little bit incremental.
>
> - I didn't see the difference between the double rounding quantization and adabit quantization.
> - The adabit quantization can also represent with [−1, 1] but not limited to [0, 1].
> - The multi-lr method is a hyper-parameter tuning, which is more like a tuning trick to me.
> - Both mixed-precision quantization and mixed-precision based on multi-bit quantization have been widely studied in previous works.
> - The improvements over Adabit are not quite significant if KD is not used.
>
> **A:** We appreciate your thoughtful analysis of our proposed multi-bit quantization framework and understand your concerns. Let us clarify the distinctions and innovations in each component of our framework:
>
> **Double Rounding vs. Adabits Quantization**
> The primary difference between Double Rounding and Adabits lies in their underlying quantization methods. Our Double Rounding quantization is an enhancement based on the LSQ[1] quantization method, where the quantization scale is learned online and not fixed. In contrast, Adabits relies on the Dorefa quantization method, where the quantization scale is determined based on the corresponding bit and calculated as $\frac{1}{2^b}$. Notably, Adabits involves normalization and rescaling operations, whereas the numerical values comparison in Figure 1 of this paper excludes these operations. Additional details on Adabits can be found [here](https://github.com/deJQK/AdaBits#centered-weight-quantization-for-low-precision). Specifically, Adabits first normalizes the continuous float values [-1,1] to [0,1], and then they will become discrete float values [0,1] after quantization and dequantization. The float values are finally restored to [-1,1] by the rescale operation.
>
> **Multi-LR Learning Rate Selection Method**
> We acknowledge that the multi-LR method is a form of hyperparameter tuning, and we advocate for its enhancement into learnable parameters. This modification aims to achieve true adaptive adjustment during training, representing a valuable avenue for further research and improvement.
>
> **Advancements Beyond Previous Works**
> While we acknowledge the extensive prior research on mixed-precision quantization and multi-bit quantization, our framework addresses specific drawbacks present in existing methods. Notably, Our Double Rounding overcomes the limitation of either only saving float models for bit switching or saving lower-bit models but with compromised accuracy.
>
> **Performance Comparison with Adabits**
> Regarding the performance comparison with Adabits without using KD, our results in Table 1 showcase competitive accuracy in the same candidate bit configuration [8, 6, 4] for MobileNetV2, despite a shorter training duration (90 epochs vs. Adabits' 150 epochs). Additionally, on ResNet18/50,  Adabits only saves the float(32bit) model due to convergence issues, while our Double Rounding method successfully saves lower-bit models, yielding superior results.
>
> We hope this detailed clarification addresses your concerns and highlights the unique contributions of our multi-bit quantization framework. Should you have any further questions or require additional information, please feel free to contact us.
>
> [1] Learned step size quantization
>
> **Q1:** Does all baseline methods use the same pre-trained model in Table-1? The full-precision baseline accuracy should be reported.
>
> **A:** Certainly, we confirm that all baseline methods in Table 1 utilize the same pre-trained model. The full-precision baseline accuracy is reported in the last column, labeled 'Float', of Table 1.
>
> Thanks again for your comment. Please feel free to contact us if further clarification is needed.

---

> > ### Author Response · Authors · 2023-11-17
> >
> > Dear anonymous reviewer,
> >
> > Thank you for your constructive comments and valuable suggestions to improve this paper. Do my responses address your concerns? If you have any questions, please feel free to contact us. We're sorry to bother you again.
> >
> > Thank you very much.
> >
> > Best regards,
> >
> > Authors

---

### Official Review · Reviewer_mFBC · 2023-11-01

**Soundness:** 2 fair
**Presentation:** 2 fair
**Contribution:** 2 fair
**Rating:** 5
**Confidence:** 4

**Summary:**

The authors present a novel approach to mixed-precision quantization, allowing for post-training bit-width selection. This method uses quantization-aware training, with the central concept being the training of a model at the highest permitted bit-width and obtaining lower precision representations through bit shifting. The authors introduce a double rounding technique that allows switching between high and low precision configurations without necessitating retraining. To address the challenges associated with simultaneously training a model for varying bit-widths, the authors advocate the use of distinct learning rates for quantization scaling parameters across different configurations, where fewer bits correspond to a smaller learning rate. The proposed method builds upon the Bit-Mixer framework (Bulat & Tzimiropoulos, 2021) with the following key differences that enhance its performance:
1. The incorporation of double rounding for efficient switching between low and high precision via bit shifting.
2. The utilization of the trace of the Hessian information during the training phase to determine the bit precision for each layer separately, with lower trace values indicating lower precision.
3. The application of different learning rates for each bit-width configuration to mitigate training instability.
4. The use of probabilities that align with Hessian information instead of employing uniform probabilities.
5. The adoption of an Integer Linear Programming (ILP) approach to determine the optimal configuration while adhering to specified constraints (e.g., FLOPs, storage).

Empirical validation on various models applied to ImageNet and CIFAR-10 datasets demonstrates the superior accuracy achieved by the proposed algorithm while using fewer or equivalent bit-widths.

**Strengths:**

1. The method allows for training models capable of dynamically adjusting their precision levels, offering adaptability for deployment on diverse edge devices.
2. Leveraging the Hessian information of each layer during bit-width assignment in the training phase enables an estimation of the number of bits required for each layer.

**Weaknesses:**

1. The introduction of additional $\mathcal{O}(n^2)$ batch normalization layers, although a minor concern, should be noted, as it may lead to additional storage costs. Nonetheless, it's worth highlighting that the size of batch normalization layers is typically smaller than that of Linear or Convolutional layers.
2. The paper could benefit from more detailed explanations of key techniques, such as the use of the Hessian trace, the precise formulation of the ILP problem, and the weighted probability method.
3. The use of Integer Linear Programming (ILP) to find the optimal bit-width configuration may be computationally intensive due to the NP-completeness of the problem. Depending on the problem size, achieving convergence to the optimal configuration may require a substantial amount of time.

**Questions:**

N/A

---

> ### Author Response · Authors · 2023-11-13
>
> We would like to thank you for your careful reading, helpful comments, and constructive suggestions, which have significantly improved the presentation of our manuscript. We have carefully addressed all comments, and below are our responses:
>
> **W1:** The introduction of additional $O(n^2)$ batch normalization layers, although a minor concern, should be noted, as it may lead to additional storage costs. Nonetheless, it's worth highlighting that the size of batch normalization layers is typically smaller than that of Linear or Convolutional layers.
>
> **A:** Thanks for the insightful observation regarding the potential additional storage costs associated with the introduction of $O(n^2)$ batch normalization (BN) layers. Firstly, we acknowledge that the incorporation of $O(n^2)$ BN layers may lead to increased storage requirements. However, this adjustment is specifically applied to the training stage of the mixed precision supernet. During this stage, the original BN is transformed from a single layer to $O(n^2)$ layers. Moreover, for the multi-bit model, the additional storage demand is reduced to $O(n)$, where $n$ represents the number of candidate bits, and $n$ does not exceed 8 at most. When transitioning from the trained supernet to the final candidate subnet for inference, the BN is seamlessly integrated into Conv layers, ensuring efficient utilization of storage space during inference. We would like to emphasize that the storage of $O(n^2)$ BN parameters only occurs when making dynamic inference decisions. Sure, it's important to highlight that the size of BN layers is smaller than that of linear or convolutional layers in the revision.
>
> **W2:** The paper could benefit from more detailed explanations of key techniques, such as the use of the Hessian trace, the precise formulation of the ILP problem, and the weighted probability method.
>
> **A:** Thanks for highlighting the need for a more detailed explanation regarding the use of Hessian traces, the precise formulation of the ILP problem, and the weighted probability methods. We appreciate the opportunity to provide additional clarification:
>
>   **Use of Hessian Trace:** We employ the Hessian trace as a sensitive metric during the training of the mixed-precision supernet and subnet decision-making. To obtain the average Hessian trace for each layer, we forward the trained full-precision model approximately 100 times through a small training set, typically comprising around 1000 images. The average Hessian trace serves as a crucial indicator for assessing the sensitivity of different layers.
>
>   **ILP Problem Formulation:** Integer Linear Programming (ILP) is mainly used to obtain candidate bits of different layers or subnet pools given constraints, such as Hessian traces, FLOPs, and parameters of candidate subnets. If directly using the heuristic greedy algorithm, it is an NP-hard problem. Our improved ILP formulation significantly simplifies the number of candidate subnet pools, thereby accelerating the optimal subnet decision-making process.
>
>   **Weighted Probability Method:** The weighted probability method is employed during the training of the mixed-precision supernet to randomly assign weights to candidate bits for each layer. The selection probability of a particular candidate bit is determined based on the previously solved Hessian trace. For instance, if we have candidate bits [8, 7, 6, 5, 4, 3, 2], more sensitive layers might be assigned weights that favor higher bits, such as [8, 7, 6, 5, 4], while less sensitive layers might lean towards lower bits like [4, 3, 2]. This process is implemented using Python code, such as `random.choices(bit_list,weights=hessian)`.
>
> We acknowledge the limitation imposed by space constraints in the main paper and hope that this more detailed explanation in conjunction with the supplementary material addresses your concerns. Please feel free to contact us for any further clarification.
>
> **W3:** The use of Integer Linear Programming (ILP) to find the optimal bit-width configuration may be computationally intensive due to the NP-completeness of the problem. Depending on the problem size, achieving convergence to the optimal configuration may require a substantial amount of time.
>
> **A:** We apologize for not explaining our improved ILP in detail. We understand the importance of addressing computational efficiency in practical applications. In fact, ILP can only solve one solution at a time for given constraint conditions, but it is basically about 1 second time by our experiments. We change the constraint conditions iteratively so that ILP can repeatedly solve multiple solutions. Basically, given a trained supernet (e.g., RestNet18), it takes less than two minutes to solve candidate subnets. Specifically, it can be implemented through the Python package `PULP_CBC_CMD`.
>
> Thanks again for your comment. Please feel free to contact us if further clarification is needed.

---

> > ### Author Response · Authors · 2023-11-17
> >
> > Dear anonymous reviewer,
> >
> > Thank you for your constructive comments and valuable suggestions to improve this paper. Do my responses address your concerns? If you have any questions, please feel free to contact us. We're sorry to bother you again.
> >
> > Thank you very much.
> >
> > Best regards,
> >
> > Authors